# Losses in the Sputum Specimen Referral Cascade in Mpulungu District, Zambia: A Cross-Sectional Study

**DOI:** 10.3390/ijerph19031621

**Published:** 2022-01-31

**Authors:** Ruth Goma, Josphat Bwembya, Brian Mwansa, Phillimon Ndubani, Francis Kasongo, William Siame, Lutinala Mulenga, Ramya Kumar, Seraphine Kaminsa, Vimbai Makwambeni, Victoria Musonda, Ibou Thior, Alwyn Mwinga

**Affiliations:** 1Ministry of Health, Mpulungu District Health Office, Mpulungu 10101, Zambia; rgoma4189@gmail.com (R.G.); briamwansa@yahoo.com (B.M.); franciskasongo17@gmail.com (F.K.); Siamewilliam71@gmail.com (W.S.); 2USAID Eradicate TB Project, PATH, Lusaka 10101, Zambia; lmulenga@path.org (L.M.); ramya.kumar.mlk@gmail.com (R.K.); skaminsa@savechildren.org (S.K.); vmakwambeni@path.org (V.M.); vmusonda@path.org (V.M.); 3Zambart, Lusaka 10101, Zambia; alwyn@zambart.org.zm; 4Frontiers Development and Research Group, Lusaka 10101, Zambia; ndubanip@gmail.com; 5PATH, Washington, DC 20001, USA; ithior@path.org

**Keywords:** tuberculosis, sputum, specimen, referral, cascade

## Abstract

Sputum specimen referral cascades in resource-limited settings are characterized by losses of specimens, resulting in delays in tuberculosis (TB) diagnosis. Mpulungu District Health Office in Zambia conducted a quantitative based cross-sectional study using both primary and secondary data to identify points at which loss of specimens occurred in the sputum referral cascade. Primary data were collected through observations and interviews with 22 TB service providers. Secondary data were collected through examination of patient files and presumptive TB and laboratory registers to retrospectively track sputum specimens referred by ten health centers from April to September 2018. Proportions of specimens/laboratory results at every stage of the referral cascade were calculated using Epi Info v7. Only 49 (23%) out of 209 sputum specimens completed the referral cascade. The remaining 160 (76%) were lost at various stages of the referral cascade. The largest loss (51%) occurred between the release of laboratory results by the diagnostic facility and their receipt at referring facilities. Barriers included an inadequate number of staff oriented in sputum specimen referral, negative staff attitudes, and lack of specimen packaging material and specimen transportation. The district health office should strengthen the sputum specimen referral system by providing transport and specimen packaging material and by training staff in sputum collection transportation and tracking.

## 1. Introduction

Tuberculosis (TB) continues to be one of the leading causes of morbidity and mortality globally. An estimated 10 million new cases and 1.6 million deaths were recorded in 2018 [1]. Zambia is among the 30 countries with the highest TB burden [1]. The National TB Prevalence Survey conducted in Zambia in 2013–2014 revealed that the prevalence of bacteriologically confirmed TB was 638 per 100,000 (502–774/100,000), with the highest prevalence recorded in the Copperbelt Province at 1211 per 100,000 (757–1665/100,000) and Lusaka Province at 932 per 100,000 (670–1195/100,000) [2].

To ensure effective control of the TB epidemic, the World Health Organization (WHO) End TB Strategy calls for early diagnosis of TB, universal drug susceptibility testing, and prompt initiation of effective treatment [3]. This collective approach can only be achieved when all patients have access to diagnostic facilities near their homes. In resource-limited settings such as Zambia, sample referral systems can ensure access to laboratory services by allowing patients to receive care and treatment at their nearest health facility, while their samples are transferred to a laboratory at a diagnostic center for testing. In this way, sample referrals increase access to diagnostic services in areas where testing is not available, which prevents the need and associated costs for patients to travel and enhances equity of access to health care [4]. According to the Zambian Ministry of Health, an ideal sputum referral cascade is characterized by seven steps [5], as presented in Figure 1.

A variety of factors negatively impact effective and efficient implementation of sample referral systems in resource-limited settings; they can range from staff to structural factors. For example, studies have attributed the inability of health care staff to collect sputum samples from presumptive TB patients at first contact to poor interpersonal communication and negative attitudes of health providers, coupled with the lack of attention and support to patients [6,7]. Other barriers include poor road networks [8] and a lack of transportation systems for TB samples, resulting in most patients being referred to the nearest diagnostic centers at their own expense [9].

In Zambia, studies have shown a lack of clear guidelines on specimen referral systems, and consequently, referral procedures vary from facility to facility [10]. There is also no clear funding mechanism in place to ensure sustainable sample referral [11]. Zambia implemented a sputum specimen courier system under the TB Care 1 project from 2010 to 2015. However, upon completion of the TB Care 1 project, the courier system ceased to function [11]. At the time of this study, the United States Agency for International Development (USAID) Eradicate TB project was in the process of supporting the National TB and Leprosy Control Program to revive the sputum specimen courier system.

Zambia has an electronic laboratory information system (DataToCare) that stores and transmits TB test results via email and SMS (short messaging system) to patients and clinicians. However, this system has yet to be completely rolled out due to operational challenges. Thus, TB services rely on delivering paper laboratory results to the referring facility rather than electronic laboratory results, which could be sent via SMS or mobile apps [12,13]. This situation results in a loss of test results along the referral cascade.

In Mpulungu District of Northern Province, routinely collected data in 2017 indicated that only three sputum specimens were referred to diagnostic facilities by all ten non-diagnostic facilities [14]. This finding was an indication of possible losses in the sputum referral cascade for Mpulungu District. This study investigated the sputum referral cascade for Mpulungu District to identify points at which losses of specimens occurred, including factors contributing to low sputum sample referrals from non-diagnostic to diagnostic centers.

## 2. Materials and Methods

### 2.1. Study Sites

The study was conducted in Mpulungu District, which is situated in the Northern Province of Zambia. Mpulungu District borders the Democratic Republic of Congo and Tanzania. The district has a population of about 130,000 [15]. The urban center is densely populated and highly involved in the fish trade, which attracts large numbers of immigrants.

At the time of this study, Mpulungu District had a total of 12 health facilities: three TB diagnostic centers and nine non-diagnostic centers. A TB diagnostic center is a health facility that is equipped with a microscope or GeneXpert used for sputum analysis to diagnose TB. Health facilities without such services are considered non-diagnostic centers. The nine non-diagnostic facilities refer sputum samples to the three diagnostic centers for laboratory confirmation of TB. Only one of the three diagnostic centers (Mpulungu District Hospital) had a GeneXpert machine. The other two diagnostic centers were only equipped with fluorescent microscopes, and in line with national TB treatment guidelines, these two diagnostic centers also refer sputum samples to Mpulungu District Hospital for diagnosis using GeneXpert. Such diagnostic centers are also referred to as ‘microscopy-only’ diagnostic centers.

Mpulungu District has poor road networks, and some areas are only accessible using fishing boats. Sputum samples are mainly transported by motorbike or boat, and in rare circumstances, a government vehicle is used. Paper-based laboratory results for patients whose sputum test positive for TB are then sent back to the referring facility via the same transport. The approximate average (median) distance to the nearest diagnostic center/GeneXpert hub for each of the non-diagnostic centers is 70 km (ranging from 5 km for Kaizya health post to 154 km for Kopeka Health Center).

### 2.2. Study Design

This was a cross-sectional study that used both primary and secondary quantitative data to investigate losses in sputum specimen referral cascade for Mpulungu District. 

### 2.3. Study Population and Sampling

All nine non-diagnostic facilities and two diagnostic centers (microscopy-only centers) that were meant to refer sputum samples to Mpulungu District Hospital were initially included in the study. Due to logistical challenges, one of the two diagnostic centers could not be reached, reducing the sample size to ten health facilities. Sputum specimens for all patients that presented to these facilities with symptoms suggestive of TB (presumptive TB patients) during the period from April to September 2018 were retrospectively tracked. All 22 health care workers that were assigned to provide TB-related services (e.g., identifying presumptive TB patients through screening, collecting, and preparing sputum samples for transport to the diagnostic center) were conveniently sampled to participate in the study.

### 2.4. Data Collection Tools and Process

This study collected data from ten health facilities (nine non-diagnostic centers and one microscopy-only diagnostic center). The data were collected by trained members of the district operational research team. The principal investigator conducted daily checks to ensure that the data collected were complete. Mentors from the USAID Eradicate TB project and Zambart also conducted data quality audits during the data collection period. An anonymized and standardized semi-structured questionnaire was administered to staff who were found providing TB services in the selected facilities, to collect data on the knowledge and attitudes of the staff. The questionnaire was administered in English. To assess levels of knowledge, staff were asked to identify signs and symptoms of TB from a list of ten options, which included the four classical signs and symptoms of TB, namely a cough lasting longer than two weeks, fever, night sweats, and weight loss. Respondents had to give a ‘yes’ or ‘no’ response to each sign or symptom. To assess staff attitudes, the questionnaire contained five statements around intensified TB case findings and sputum specimen collection and referral. Respondents needed to give responses ranging from ‘strongly agree’ to ‘strongly disagree’. In the last section of the questionnaire, respondents were asked to indicate ‘yes’ or ‘no’ to each item on the list of eight possible structural-related barriers to sputum sample referral, which among others, included lack of commodities needed to collect sputum specimens, a lack of transport, and high workloads. Additional primary data were obtained using a checklist on availability of material and equipment necessary for sputum specimen collection, packaging, and transportation. 

To identify points of losses in the sputum referral cascade, a data extraction form bearing patient identifiers (i.e., names, age, sex, address, and file numbers) was used to retrospectively track sputum samples for patients who presented with signs and symptoms of TB at non-diagnostic sites in the period between April and September 2018, from outpatient and inpatient medical records to the presumptive TB register, to the laboratory specimen referral form at non-diagnostic sites, to the laboratory register at the diagnostic center, and back to the presumptive TB register at the referring non-diagnostic site. The data extraction form also collected dates for sample collection, sample receipts at diagnostic sites, and diagnostic result receipts at the referring non-diagnostic facility. These dates were used to calculate the turnaround time (TAT) from when the sputum sample was collected from the patient to when the referring non-diagnostic facility received the diagnostic results.

### 2.5. Data Management and Analysis

Both primary and secondary data were entered into Epi Info v7. We conducted data quality checks and then ran frequency distributions to describe the demographic characteristics, training in sample referral, knowledge on intensified TB case finding, attitude toward sample referral, and perceived challenges to sputum sample referral. To establish levels of knowledge, each correct response to the questions on signs and symptoms of TB was assigned a score of one. A score of zero was assigned to a wrong response. Average scores for each respondent were then calculated and categorized as low (average score < 0.5), moderate (average score between 0.5 and 0.74) or high (average score between of 0.75 and above).

To establish the points of losses in the sputum specimen referral cascade, we calculated: (1) the proportion of TB presumptive patients that submitted their sputum to non-diagnostic sites in the period between April and September 2018; (2) the proportion of sputum specimens that were referred to diagnostic sites for testing; (3) the proportion of referred sputum specimens that were received at diagnostic centers; (4) the proportion of received sputum specimens that were examined at the diagnostic facility; (5) the proportion of examined sputum specimens whose results were sent back to referring centers; and (6) the proportion of sputum specimen results released from diagnostic centers that were received at referring facilities. 

## 3. Results

### 3.1. Analysis of the Sputum Specimen Referral Cascade

Out of the 307 presumptive TB patients (i.e., patients presenting with one or more symptoms suggestive of TB), 209 (68%) submitted sputum for diagnosis (Figure 2). Of the 209 samples submitted to non-diagnostic centers, 205 (98%) of them were referred to diagnostic centers. Of the 205 samples referred to diagnostic centers, 149 (73%) of them were received at diagnostic centers. Of the 149 samples received at diagnostic centers, 146 (98%) of them were examined. Of the 146 samples examined in the laboratory, 100 (68%) had their results sent back to referring facilities. Of the 100 samples whose results were dispatched from diagnostic centers, 49 (49%) of them were received at a referring facility. As a result of losses at every stage, only 24% of sputum specimens were successfully moved from non-diagnostic centers to the diagnostic center and results received back at the non-diagnostic centers to complete the referral cascade. The largest loss occurred between the dispatch of results from diagnostic centers and their receipt at non-diagnostic centers.

The mean total TAT for the 49 samples that successfully completed the referral cascade was four days (SD = 3.4). The longest delay occurred between sample dispatch and sample receipt at the diagnostic facility (mean = two days; SD = three days).

### 3.2. Barriers to Sputum Sample Referral in Mpulungu District

This study characterized the barriers to the sputum referral system in Mpulungu District into two categories: staff and structural barriers.

#### 3.2.1. Staff-Related Barrier to Sputum Sample Referrals in Mpulungu District

Among staff-related factors, the study found that the majority (77.3%) of the 22 staff providing TB services in the facilities visited fell into the category of ‘other’ (Table 1). This category included non-clinical staff such as community health assistants and classified daily employees. The majority (68.2%) had worked for less than two years. Of the staff who had not received formal orientation on sputum sample collection and referral standard operating procedures (SOPs), 77.3% had moderate to high levels of knowledge of the signs and symptoms of TB. 

Regarding staff attitude, the majority (77.2%) of the respondents either agreed or strongly agreed with the statement that ‘health facility staff need to promptly identify patients with TB’, and 40.9% also agreed or strongly agreed with the statement that ‘presumptive TB patients are difficult people to deal with’. In addition, 27.2% of staff either agreed or strongly agreed with the statement that ‘TB sputum sample collection and packaging is an unpleasant task’ (Table 2).

#### 3.2.2. Structural-Related Barriers to Sputum Sample Referral

Lack of transport to courier sputum specimens was the most commonly cited structural barrier, with over 70% of the respondents listing this as a challenge. This was followed by a lack of supplies necessary for sputum collection, packaging, and transportation (45.5%) and heavy workloads (13.6%). Respondents were then asked to indicate the most commonly used mode of specimen transportation; public transport and motorbikes were the most common modes of specimen transportation cited by 40.9% of the respondents. A physical check (using a separate checklist) also revealed that only half of the facilities had their own mode of transport (motorbikes), while the remaining half did not.

We also assessed whether or not basic commodities needed for collection and packaging of sputum specimens were available. Table 3 shows that most facilities had the basic commodities needed to collect sputum specimens. However, none of the facilities had Ziplock bags, which were necessary for the triple packaging system to safely refer sputum samples, and only two (20%) of the health facilities has standard laboratory request forms.

## 4. Discussion

Our results show that there were losses of every stage of the referral cascade. First, sputum specimens were not collected from 32% of presumptive TB patients surveyed that presented to the facilities. This may be attributed to failure to produce sputum and fear of being diagnosed with TB (due to stigma) on the part of presumptive TB patients. Lack of means for transporting specimens to diagnostic centers may also discourage health care workers from collecting sputum from patients.

This study also found that of the sputum specimens collected from patients, only 24% successfully completed the referral cascade. This figure was far lower than what was recently found in a rural district in Zimbabwe, where 72% of samples collected managed to complete the referral cascade [16]. It was, however, higher than what was found in an earlier study conducted in Tanzania where only 9% of samples managed to complete the referral cascade [17]. 

These findings have far-reaching public health implications. The failure to collect sputum from presumptive TB patients, examine it, and deliver test results poses a serious risk to the community, as some individuals may be carrying active TB, which they will continue to spread. It is estimated that a person with infectious TB can infect 10–15 other people in a year [16]. Further, a delay in diagnosing TB (due to delayed sputum collection), means delayed treatment initiation, which is associated with a higher risk of adverse outcomes such as death [18]. It is therefore important that strategies are put into place to ensure that all presumptive TB patients submit sputum; thus, those with active TB are initiated on treatment to prevent further transmission and mortality. Patient education on coughing techniques is one way of increasing the number of presumptive TB patients that submit sputum for examination. 

There is also a need for specimens collected from patients to reach the test laboratory and for results sent back to the referring health facility. One way to do this is by introducing a sputum tracking system that will enable staff at both referring and testing centers to visualize samples and results in transit. In Lesotho, the Clinton Health Access Initiative developed a mobile sample tracking system application on ODK Collect, which was installed on mobile phones for health care workers and motorbike riders. The mobile application consisted of four data capturing applications, each documenting a stage in the sample referral cascade: sample pickup, sample delivery, results pickup, and results delivery. The project used barcode stickers with unique number sequences that could be affixed to samples, forms, and registers. Mobile phones for motorbike riders were used as barcode scanners. Information from the mobile application was transmitted to the sample transport tracking dashboard, which served as a comprehensive tool used to display data, including visualization of data at national, district, and facility levels [4]. 

Maintaining a short TAT for sputum referred for examination is another way of ensuring early treatment and better outcomes for TB patients [4]. This study found a relatively lower TAT (median = four days) that was comparable to what was found in a similar study conducted on Tonga Island [19] and lower than the seven days recorded in Zimbabwe [16]. However, the reliability of TAT calculations in our study was negatively affected by the small number of samples that completed the referral cascade. Missing dates at various stages of the cascade also reduced the number of samples included in the calculation of TAT. Our findings may therefore not be a true representation of the situation in Mpulungu District. 

Barriers to sputum specimen referral in this study were divided into staff and non-staff related. The first staff-related barrier identified was the high number of non-clinical staff providing TB services, a majority of whom had not been oriented in sputum specimen referral SOPs. Use of non-clinical staff has two major limitations. First, their ability to identify presumptive TB patients required to submit sputum may be low. Second, non-clinical staff may be less likely to apply and follow biosafety measures during sputum sample collection and transportation compared to clinical staff. There is therefore an urgent need for authorities in Mpulungu District to orient staff in sputum referral SOPs. In Uganda, training of staff in sputum referral SOPs, among other interventions, increased the number of sputum specimen referrals by more than tenfold, with 94% of specimens reaching the testing laboratory [8].

The second staff-related barrier identified by this study was the negative staff attitudes toward presumptive TB patients. Over 40% of the staff interviewed either ‘agreed’ or ‘strongly agreed’ with the statement that ‘TB presumptive cases are difficult people to deal with.’ This negative attitude may compromise staff abilities to counsel and persuade presumptive TB patients to submit sputum. This finding may, in part, explain the high number of presumptive TB patients who did not submit sputum. 

Among the non-staff-related barriers identified by this study was the shortage of Ziploc bags and standard sputum containers that are necessary for the triple packaging system recommended by WHO [4]. The triple packaging technique ensures that the samples arrive at the diagnostic laboratory in ideal conditions for testing [20]. Additionally, the majority (80%) of the facilities surveyed by this study did not have standard laboratory request forms, which may result in missing vital information on referred samples. Specimens lacking vital information may be rejected by the testing laboratory, thus delaying TB diagnosis. Although almost all the specimens that reached the laboratory in this study were tested, lack of packaging material and a standard referral form is an important gap that needs to be addressed. 

Reliable transport is critical to the successful completion of the sputum referral cascade. Half of the health facilities surveyed by this study did not have any mode of transporting sputum samples. This challenge has also been reported in other resource-limited settings [16,21]. Lack of transport at the health facility may lead to the staff resorting to use of public transport, resulting in delays, compromised specimen quality, and loss of test results. Innovative strategies such as use of motorbike riders to pick up specimens and deliver test results can help resolve transportation challenges. The strategy has proved effective in a number of resource-limited countries, including Uganda [8], Mozambique, Nigeria, and Zimbabwe [4].

The major strength of our study was the ability to track sputum specimens along the entire referral cascade, while at the same time identifying barriers on the way to successful completion. This approach helped identify areas of improvement in the sputum referral cascade. Limitations included missing data, such as dates in secondary data sources, which were required in the calculation of TAT. Missing dates reduced the number of sputum specimens included in the calculation TAT, hence reducing the reliability of our findings in this area. In addition, due to logistical (transport) challenges, we could not collect data from one health facility, reducing the sample size to ten health facilities. Notwithstanding these limitations, our study brings out vital information that could help improve the sputum referral systems in Mpulungu District and beyond.

## 5. Conclusions

This study investigated the sputum referral cascade for Mpulungu District in Zambia. Findings indicate that there were losses at every stage of the referral cascade. These losses could in part account for low sputum sample referrals recorded in the district. Barriers in the sputum specimen referral cascade for Mpulungu District included a lack of SOPs and transport facilities, use of non-clinical/nursing staff who are not oriented in sample referral, negative staff attitude, and limited sputum sample packaging materials. These findings call for a provision of SOPs and job aids on sputum sample collection, packaging, transportation, and tracking to all health facilities in Mpulungu District. Staff involved in provisional TB services must be oriented in use these tools. Moreover, there is a need for engagement with partner projects to leverage transportation using motorbikes and piggybacking other disease sample transportation systems.

## Figures and Tables

**Figure 1 ijerph-19-01621-f001:**
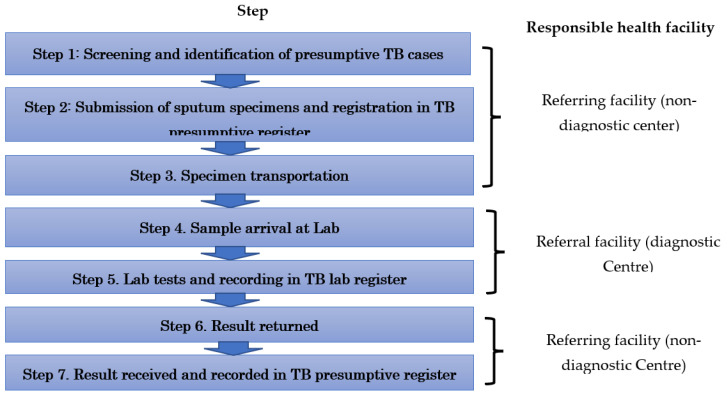
Steps in an ideal sputum specimen referral cascade.

**Figure 2 ijerph-19-01621-f002:**
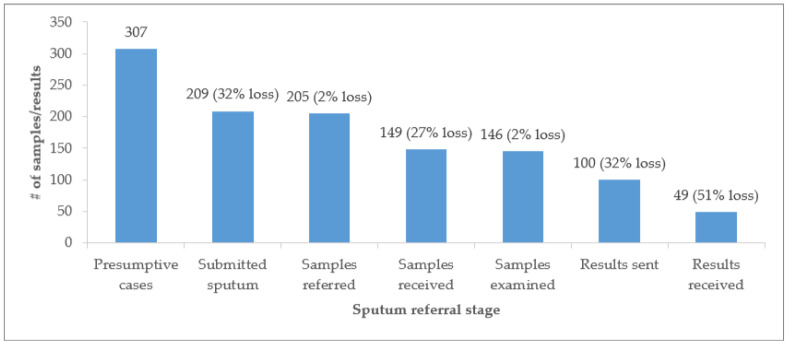
Sputum sample referral cascade for patients presenting with symptoms suggestive of TB in ten public health facilities in Mpulungu District (April–September 2018).

**Table 1 ijerph-19-01621-t001:** Characteristics of TB service providers from ten non-diagnostic facilities in Mpulungu District, October 2018 (*N* = 22).

Variable	Frequency	Percentage
Sex		
Male	14	63.6
Female	8	36.4
Age (in years)		
20–30	16	72.7
31–40	3	13.6
41–50	2	9.1
Above 50	1	4.5
Health profession		
Clinical officer	2	9.1
Nurse	2	9.1
Laboratory staff	1	4.5
Others	17	77.3
Period in service (in years)		
Less than 2	15	68.2
3–5	3	13.6
6 and above	4	18.2
Knowledge on TB signs and symptoms		
High	10	45.5
Moderate	7	31.8
Low	5	22.7
Oriented to sputum specimen referral SOPs		
Yes	5	22.7
No	17	77.3

**Table 2 ijerph-19-01621-t002:** Attitudes toward intensified TB case finding among TB service providers from ten non-diagnostic facilities in Mpulungu District, October 2018 (*N* = 22).

Statement	Strongly Disagree (%)	Disagree (%)	Neutral (%)	Agree (%)	Strongly Agree (%)
Health facility staff need to promptly identify patients with TB.	15.6	4.5	4.5	63.6	13.6
TB presumptive patients are difficult people to deal with.	9.1	50.0	0.0	27.3	13.6
TB sputum sample collection and packaging is an unpleasant task.	4.5	54.5	13.6	22.7	4.5
There is nothing we can do about undetected TB presumptive patients because it is not our fault.	40.9	36.4	4.5	4.5	13.6
There is much we can do to send more sputum samples to the diagnostic centers.	4.5	9.1	0.0	40.9	45.5

**Table 3 ijerph-19-01621-t003:** Availability of material/equipment for collection and packaging of sputum specimens in ten non-diagnostic centers in Mpulungu District (October 2018).

Type of Commodity	Number of Facilities Where Available (%)
Absorbent material	9 (90)
Adhesive tape	10 (100)
Lab request form	2 (20)
Plastic Ziplock bags	0 (0)
Cooler boxes	10 (100)
Gloves	10 (100)
Sputum containers	7 (70)

## Data Availability

The data presented in this study are available from the corresponding author on reasonable request.

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
