# Peer review of "Losses in the Sputum Specimen Referral Cascade in Mpulungu District, Zambia: A Cross-Sectional Study"

_ijerph, 2022, doi:10.3390/ijerph19031621_

Round 1

Reviewer 1 Report

Journal: International Journal of Environmental Research and Public Health

Title: Losses in the Sputum Specimen Referral Cascade in Mpulungu District, Zambia: A Cross-Sectional Study

Manuscript Number: ijerph-1547194

This study is a cross-sectional study exploring the losses in the sputum specimen referral cascade in Mpulungu District, Zambia. The challenging nature of curbing the spread of TB and obtaining TB tests in remote areas of Africa is addressed.

The manuscript was clear, concise, and easy to understand.  The study is very simple but quite informative.  An evident need was described in the introduction, straightforward parameters in the methods with easy-to-understand results. The only main item would like to see more information on is the questionnaires themselves, who administered them, etc. As the questionnaires are semi-formal, I assume aren’t standardized or validated. Are they in English/local dialect/any translation issues/any issues with culture or customs? Where the questions structured in a way to reduce bias? Who is conducting the questionnaires, perhaps more information about the “district operational team”?

The discussion provides comparisons to similar studies and recommendations to improve public health practices.  The research is not quite novel, as referenced in the paper, because similar research was performed in other areas of Africa. However, this paper does add to the documentation of the deficiencies in the sputum referral system and establishment of potential solutions to help reduce TB transmission and morbidity. The paper addresses one of its major weaknesses, the lack of data, numbers of records and participating facilities, and acknowledges that their findings might not be a true representation of the situation in Mpulungu District.  Tables and figures seem appropriate and added a visual representation of the results.  No in-depth statistical analyses were performed most likely due to the small number size.  The conclusions were consistent with the evidence and arguments presented. I found no need for major revisions. Overall, the presentation of the information was very-well done.

Reviewer 2 Report

The paper is well written and coherent to the aim of the journal (Journal of Environmental Research and Public Health) and it is interesting for the international reader to have a look in to a particular scenario such as the Zambian one.

Affiliation

the email address of each author should be added and the institution should be written according to the guidelines of the journal.

Introduction

well written and easy to read and understand the endpoints.

Figure 1 should be shaper.

Materials and Method

this section is written according to the guidelines for the authors and it is appropriate.

Results Discussion and Conclusion 

I have no remarks on these section.

The paragraph of the author contribution should be written according to the template example
